# A Novel Approach for Investigating Upper Airway Hyperresponsiveness Using Micro-CT in Eosinophilic Upper Airway Inflammation such as Allergic Rhinitis Model

**DOI:** 10.3390/biom9070252

**Published:** 2019-06-27

**Authors:** Dan Van Bui, Akira Kanda, Yoshiki Kobayashi, Yoshiko Sakata, Yumiko Kono, Yoshiyuki Kamakura, Takao Jinno, Yasutaka Yun, Kensuke Suzuki, Shunsuke Sawada, Mikiya Asako, Akihiko Nakamura, David Dombrowicz, Keita Utsunomiya, Tanigawa Noboru, Koichi Tomoda, Hiroshi Iwai

**Affiliations:** 1Department of Otorhinolaryngology, Head and Neck Surgery, Kansai Medical University, Hirakata 573-1010, Japan; 2Allergy Center, Kansai Medical University Hospital, Hirakata 573-1010, Japan; 3Central Research Laboratory, Kansai Medical University, Hirakata 573-1010, Japan; 4Department of Radiology, Kansai Medical University, Hirakata 573-1010, Japan; 5Department of Information Systems, Faculty of Information Science and Technology, Osaka Institute of Technology, Hirakata 573-0196, Japan; 6Nakamura ENT Clinic, Sakai 591-8025, Japan; 7University of Lille, Inserm, CHU Lille, Institut Pasteur de Lille, U1011-EGID, F-5900 Lille, France

**Keywords:** Allergic rhinitis, airway hyperresponsiveness (AHR), eosinophilic airway inflammation, micro-CT, and nasal resistance

## Abstract

Airway hyperresponsiveness (AHR) has been proposed as a feature of pathogenesis of eosinophilic upper airway inflammation such as allergic rhinitis (AR). The measurement system for upper AHR (_U_AHR) in rodents is poorly developed, although measurements of nasal resistance have been reported. Here we assessed UAHR by direct measurement of swelling of the nasal mucosa induced by intranasal methacholine (MCh) using micro-computed tomography (micro-CT). Micro-CT analysis was performed in both naïve and ovalbumin-induced AR mice following intranasal administration of MCh. The nasal cavity was segmented into two-dimensional horizontal and axial planes, and the data for nasal mucosa were acquired for the region of interest threshold. Then, a ratio between the nasal mucosa area and nasal cavity area was calculated as nasal mucosa index. Using our novel method, nasal cavity structure was clearly identified on micro-CT, and dose-dependent increased swelling of the nasal mucosa was observed upon MCh treatment. Moreover, the nasal mucosa index was significantly increased in AR mice compared to controls following MCh treatment, while ovalbumin administration did not affect swelling of the nasal mucosa in either group. This UAHR following MCh treatment was completely reversed by pretreatment with glucocorticoids. This novel approach using micro-CT for investigating UAHR reflects a precise assessment system for swelling of the nasal mucosa following MCh treatment; it not only sheds light on the mechanism of AR but also contributes to the development of new therapeutic drugs in AR patients.

## 1. Introduction

Allergic rhinitis (AR) is a type 2 T helper cell (Th2)-skewed disease accompanied by eosinophils infiltration into nasal tissue, following exposure to inhaled antigens in the upper airway in genetically predisposed individuals. The prevalence of AR has been increasing, and over 500 million people suffer from AR worldwide [1]. Allergic rhinitis is diagnosed through a typical history of three characteristic symptoms (sneezing and nasal scratching, water rhinorrhea, or nasal obstruction) and clinical examination findings: detection of eosinophils in nasal discharge, nasal provocation test using antigen, elevation of allergen-specific immunoglobulin E (IgE) antibodies, or skin prick test [1,2]. Likewise, asthma is diagnosed on the basis of clinical symptoms (wheezing and dyspnea) and clinical examination findings: index of airway responsiveness in the respiratory tract to detect airway hyperresponsiveness (AHR), bronchodilator test, exhaled nitric oxide concentrations, and/or increased serum levels of specific IgE [3].

AHR is an exaggerated response to a variety of irritants including histamine, methacholine, bradykinin, isotonic or hypertonic solutions, and cold or dry air; it is one of the hallmark features of allergic airway inflammation [4,5,6]. The mechanisms of AHR are implicated in multiple changes in the airway wall, characterized by epithelial cell layer disruption, increased collagen deposition, hypertrophy of smooth muscle, and increased vascularization [7]. Sensory nerves and endings of the parasympathetic nervous system become exposed; this leads to increased sensitivity to endogenous or exogenous stimuli, causing persistence of airway inflammation [8]. Clinically, lower AHR (_L_AHR) in asthma is reflected by an increased airway resistance due to constriction of the bronchial smooth muscles following nonspecific stimulants such as methacholine (MCh) or histamine. Clinical examination for _L_AHR through nonspecific stimulant provocation test is routinely performed to diagnose and assess the severity of asthma in clinical practice. Conversely, upper AHR (_U_AHR) in AR is usually evaluated on the basis of onset of clinical symptoms (nasal obstruction, nasal secretion and sneezing) after provocation. However, measurement of _U_AHR in AR by which indicated as nasal resistance following provocation is rarely performed in clinical practice, although clinical trials using nasal peak flow or rhinomanometry have been performed [1,9,10].

In animals, to detect AHR following MCh administration in the animal model, measurement of _L_AHR, reflected by lung resistance and compliance, is commonly performed using non-invasive or invasive plethysmography [11,12]. Animals are evaluated 24 to 48 h after the last allergen challenge because the difference between naïve and asthma groups in _L_AHR the highest during late-phase inflammation following the last antigen challenge [13,14]. In contrast, no studies have reported the evaluation of _U_AHR following MCh administration in the upper airway eosinophilic inflammation mimicking AR, while evaluation in the early phase by specific antigen has been performed by using frequency of sneezing and scratching. Although mechanism of _U_AHR is unclear, threshold elevation by MCh in AR model indicates hyperresponsiveness. Despite the demand for a _U_AHR evaluation system that can directly measure swelling of the nasal mucosa rather than nasal resistance, no measurement system has been developed.

Recently, micro-computed tomography (micro-CT) has been widely used for analysis of bone and soft tissue in animals [15]; it can provide high spatial and temporal resolution images [16]. Analysis using micro-CT is an innovative method for evaluation of anatomical structures and pathological lesions compared with histological analysis [17]. Here we investigated the application of micro-CT for _U_AHR evaluation to measure swelling of the nasal mucosa following MCh intranasal administration in an AR mouse model and assessment of anti-inflammation compounds.

## 2. Methods

### 2.1. Mice

BALB/c mice (weight 20 g, selected from female 6–8 weeks), purchased from Shimizu Experimental Material (Kyoto, Japan), were bred in a specific pathogen-free animal facility with a regular 12 h light/dark cycle. All experiments were performed in accordance with the Animal Care and Use Committee of Kansai Medical University (18-082).

### 2.2. Ovalbumin (OVA)-Induced AR Model

Protocol of schema is shown in Appendix A. Briefly, mice were sensitized through intraperitoneal injection of 50 μg OVA (Sigma-Aldrich, Missouri, USA.) or phosphate buffered saline (PBS) with 1 mg of aluminum hydroxide (Thermo Fisher Scientific, Massachusetts, USA) on Days 0 and 14 and daily challenged with OVA (nebulization of 1% OVA for 30 min) during Days 21–25. On Day 26, micro-CT analysis was performed 24 h after the last challenge. Then, OVA-induced AR mouse model was prepared (see histology in Appendix A). In some experiments, mice were pretreated with an intraperitoneal injection of dexamethasone (10 mg/kg, Sigma-Aldrich) or sesame oil with 4% dimethyl sulfoxide as vehicle control 1 h before each OVA challenge on Days 21, 23 and 25.

### 2.3. Evaluation of Nasal Mucosa by Micro-Computed Tomography

Mice were treated with 2 μL of PBS or 0.5–2.0 mg/mL MCh (Sigma-Aldrich) that was administered into each nostril for 15 min under systemic anesthesia with a mixture of 50 mg/kg ketamine (Ketalar, Daiichi-Sankyo, Tokyo, Japan) and 0.5 mg/kg medetomidine (Domitor; Pfizer, New York City, USA). In the study of time course, mice were treated with 2μl MCh (1mg/mL) or PBS intranasal administration for 0, 15, 60 or 120 min under anesthesia. Then, mice were sacrificed with an intraperitoneal injection of pentobarbital (Kyoritsu Pharmaceutical, Nara, Japan), and micro-CT imaging was immediately performed using the specifically on single photon emission computed tomography/computed tomography (SPECT/CT) system (Siemens, Bayern, Germany) with the following scanner settings: tube voltage of 70 kVp and current of 500 μA over 360 continuous projections with an exposure time of 1000 ms per projection. An aluminum filter (0.5 mm) was used to reduce beam-hardening artifacts. The cross-sectional images were reconstructed using Inveon Viewer QuickLaunch software (Siemens Healthcare, Erlangen, Germany) and converted to the Digital Imaging and Communications in Medicine format using PMOD software version 3.703 (PMOD Technologies, Zurich, Switzerland). For each mouse, a stack of 768 cross-sections was reconstructed, and manual regions of interest (ROIs) were evaluated using OsiriX software version 8.0.1 (Pixmeo SARL, Bernex, Switzerland). Data were reconstructed using Inveon Viewer QuickLaunch software version 4.2.0.15. The reconstructed two-dimensional (2D) images (1088 × 1088 pixels) comprised 19.71 μm voxels with 10,000 Hounsfield Units (HU) (window width) and 2000 HU (window level).

### 2.4. Statistical Analysis

Data are presented as means ± standard errors of mean (SEMs). Statistical significance was determined using Mann–Whitney U test by software GraphPad Prism version 7 (GraphPad, San Diego, USA). The threshold of significance was set at *p* < 0.05 for all tests.

A supplementary methods section can be found in this article’s online repository.

## 3. Results

### 3.1. Establishment of Quantitative Analysis for Nasal Mucosa Using Micro-Computed Tomography

To develop a novel approach for measurement of nasal mucosa swelling, we first defined the analysis zone within the nasal cavity (see 2D sagittal plane in Figure 1A); the lowest slice was determined by connecting the anterior tip of the nasal bone and the anterior arch of atlas; the highest slice on the top of horizontally segmented seven slices from the lowest level until the middle of the nasal bone. To completely remove the oral cavity area from the analysis, the first two slices from the lowest section were excluded. Then, a total of five images were used for analysis (slices from A1 to A5, as seen in Figure 1A). The nasal mucosa area was automatically recognized using a bi-threshold approach (−700 to +700 HU), while the nasal cavity area was manually selected (Figure 1B,E). The mucosa index was calculated as the mean of arbitrary values: mucosal area/total area in five slices (Figure 1C,F).

For analysis of the coronal sections, the most anterior slice was determined by connecting the root of the upper incisor tooth and the sagittal suture on sagittal plane (2D sagittal plane seen in Figure 1D). The slice of the posterior end was indicated by segmentation of five slices connecting the sagittal suture and the joint between the frontal and parietal bones (slices from C1 to C5 seen in Figure 1D)

### 3.2. Measurement of MCh-Induced Swelling of the Nasal Mucosa

To investigate the effect of MCh on the nasal mucosa in naïve mice, PBS or 1 mg/mL MCh was intranasally administered 15 min before micro-CT analysis. As expected, MCh-induced swelling of the nasal mucosa compared with the PBS-treated group (Figure 2A). Moreover, we found that the mucosa index increased with MCh treatment in a dose-dependent manner; PBS, 0.5, 1.0, and 2.0 mg/mL MCh intranasal administration (Figure 2B). There were significant differences between PBS alone- and MCh-treated groups regarding the increase in the mucosa index, and this increase reached a plateau at 2.0 mg/mL MCh.

In kinetics analysis, a significant difference from baseline was found 15 min and 60 min after 1 mg/mL MCh treatment, whereas no significant difference was observed 120 min after the treatment (Figure 2C).

Together, these data suggest that MCh is most effective at a concentration of 1.0 mg/mL 15 min after administration. Moreover, similar results were observed in both dose and kinetic analyses in the coronal slice analysis (Appendix A).

### 3.3. Increasing in the Mucosa Index in Allergic Rhinitis

To investigate whether _U_AHR was significantly induced in the AR mice model, these mice were treated with PBS or MCh 24 h after the last OVA challenge. As shown in Figure 3A, swelling of the nasal mucosa was clearly observed in the AR group compared with that in the control group, following treatment with 1 mg/mL MCh. Regarding mucosa index with the 0.5 and 1.0 mg/mL MCh treatments, there were significant differences between the AR and control groups (Figure 3B), while no significant difference was observed with PBS and 2.0 mg/mL MCh treatment. Moreover, to investigate whether OVA treatment following repeated OVA provocation induces swelling of the nasal mucosa, mucosa indexes were analyzed 15 min after OVA treatment 24 h following repeated OVA challenges. As shown in Appendix A, OVA treatment did not affect the increase in swelling of the nasal mucosa in both the AR and control groups; furthermore, no significant differences were observed between the AR and control groups after treatment with 1 mg/mL MCh. Therefore, these data indicate that OVA treatment does not alter swelling of the nasal mucosa.

### 3.4. Steroids Reduced Increasing of Upper Airway Hyperresponsiveness in Allergic Rhinitis

Because glucocorticoids have a strong potential to suppress an allergic response with AHR inhibition, mice were pretreated with dexamethasone via systemic administration. The increase in MCh-induced mucosa index in AR was completely abrogated by dexamethasone pretreatment (Figure 4).

## 4. Discussion

We found that the structures of the nasal cavity and the paranasal sinuses could be clearly identified on 2D reconstructed images using an image analyzer for animals and a micro-CT scanner. Although studies have reported the use of micro-CT to determine the anatomy of the nasal cavity and paranasal sinuses [18,19], there have been no reports of a quantitative approach to measure the degree of swelling of the nasal mucosa. In this study, we initially developed a procedure involving the measurement area and condition, followed by the development of a quantitative method focused on measuring swelling of the nasal mucosa using mucosa index. Regarding _U_AHR evaluation, we showed an increase in _U_AHR in an AR mouse model due to MCh-induced swelling of the nasal mucosa using a micro-CT scanner and suggested an original method for _U_AHR evaluation.

A functional in vivo assay for an AR mouse model was set up by measuring the frequency of sneezing and nose-scratching for 10 min after the last challenge with specific antigen or nonspecific antigen [20,21,22]. However, this assay can only detect early-phase response, not late-phase responses and nasal resistance, which is one of the biological responses following AHR. Increasing _L_AHR in asthma patients reflects tissue remodeling with destruction of the epithelial layer, especially in those with persistent inflammation [7]. In an AR patient, _U_AHR is also enhanced by repeated nasal challenges with specific antigens [6] and is involved in persistence of inflammation. Similar results have been reported in an AR animal model [23]. However, airway resistance in asthma and AR results from the contraction of the bronchial smooth muscles and swelling of the nasal mucosa with nasal discharge, respectively [24,25]. Unlike the evaluation of airway resistance in asthma, evaluation of nasal resistance is complicated because it requires consideration of not only this different pathway but also the influence of the lower airway. Therefore, few studies have focused on evaluation using plethysmography in an AR model, whereas this approach has been widely used to determine the underlying mechanism or to evaluate the effect of new therapeutic compounds in an antigen-induced asthma model [26].

Regarding previous studies using airway functional tests indicated for nasal airway resistance or respiratory frequency accompanied with resistance, Miyahara et al., using non-invasive measures such as whole-body plethysmography or invasive plethysmography, reported that both allergen-specific and nonspecific provocations were involved in _U_AHR of early- and late-phase nasal responses [23,27]. Likewise, Mizutani et al. reported nasal hyperresponsiveness to histamine, but not to MCh, induced by repetitive inhalations of cedar pollen into the nose of guinea-pigs [28]. However, it is impossible to block the background factors such as the effect of the lower airway, nasal discharge, and body motion of mouse, and induction of AR in non-asthmatic animals has not been achieved because of technical challenges and interactions between upper and lower airway diseases through drainage of mediators, neural reflex via nervous vagus, and/or systemic dissemination of mediators [29]. The evaluation of _U_AHR using invasive plethysmography for upper airway is affected by surgical stress and anesthesia but not by the lower airway [30]. Taken together, currently, measurement of nasal resistance using whole-body plethysmography or invasive plethysmography does not seem to be an appropriate approach to evaluate _U_AHR.

In contrast, using the approach involving measurement of swelling of the nasal mucosa, we could detect an _U_AHR-induced direct effect (see schema in Appendix A). This system could assess reactivity of MCh 24 h after the last OVA challenge, as reflected in _U_AHR in the late-phase nasal responses. Notably, we could not find any increase in the mucosa index by OVA treatment 24 h following repeated OVA challenges, suggesting that the increased nasal resistance and respiratory frequency reported in previous studies might reflect nasal discharge [23,27]. Moreover, we found abrogation of _U_AHR with steroids, which indicated that swelling of the nasal mucosa was induced by _U_AHR through epithelial layer destruction. However, innovation of an apparatus and cost reduction is required because micro-CT requires expensive equipment, and a long time (almost 20 min/mouse) is required to capture high-resolution images.

In conclusion, our novel approach of evaluation of _U_AHR involving measurement of swelling of the nasal mucosa after MCh pretreatment using micro-CT not only sheds light on the potential mechanisms underlying AR but will also help with the development of a new therapeutic drug in an AR mouse model.

## Figures and Tables

**Figure 1 biomolecules-09-00252-f001:**
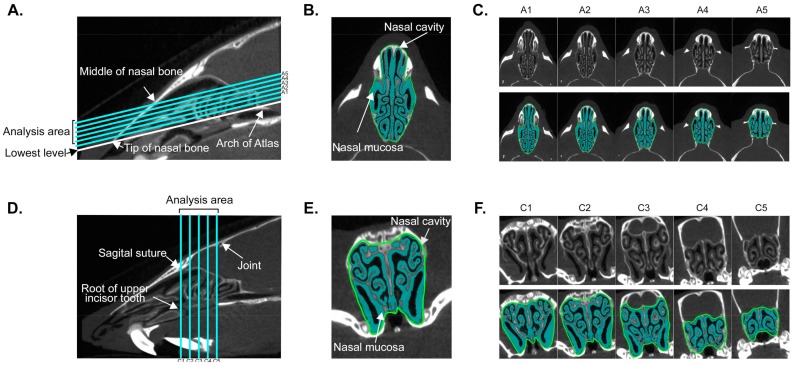
Landmark definition in micro-computed tomography (CT) imaging. **A** and **D** indicate segmentation of the axial and coronal axes on sagittal CT images of nasal cavity, respectively. **B** and **E** show the region of interest (ROI) on slice A2 in axial and C2 in coronal axis, respectively. **C** and **F** indicate serial CT images in raw data and ROI on axial (A1–A5) and coronal (C1–C5) axes, respectively.

**Figure 2 biomolecules-09-00252-f002:**
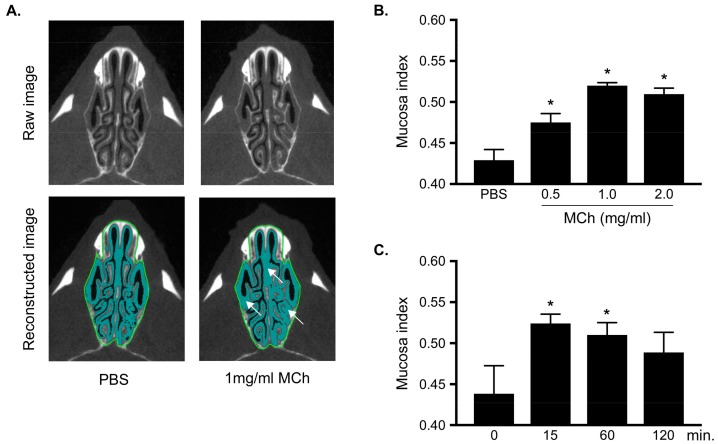
Analysis of swelling of nasal mucosa using micro-CT images in axial axis (**A**), Upper and Lower panels indicate raw and reconstructed selected ROI images, respectively. Left and right panels indicate observations 15 min after PBS and 1 mg/mL methacholine (MCh) treatments, respectively. (**B**) and (**C**) show dose-dependent and kinetics analyses, respectively. Mucosa index was analyzed 15 min after PBS or 0.5–2.0 mg/mL MCh treatment (**B**) or 0, 15, 60, and 120 min after PBS or 1 mg/mL MCh treatment (**C**). Data are expressed as means ± standard errors of mean (SEMs) (*n* = 3–6 for each group). * Significantly different comparing with control group (*P* < 0.05).

**Figure 3 biomolecules-09-00252-f003:**
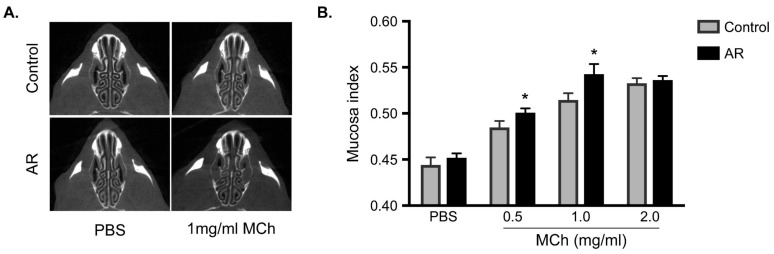
Increase in the mucosa index in OVA-induced allergic rhinitis (AR). (**A**) indicates raw axial micro-CT image at 15 min after PBS or 1 mg/mL MCh treatment in control (upper panel) and OVA-induced AR group (lower panel). (**B**) Mucosa index is shown 15 min after PBS or 0.5–2 mg/mL MCh treatment in control (gray bar) and AR (black bar) groups. Data are expressed as means ± SEMs (*n* = 5 for each group). * Significantly different from control mice (*P* < 0.05). Increase in the mucosa index in OVA-induced allergic rhinitis (AR). (**A**) indicates raw axial micro-CT image at 15 min after PBS or 1 mg/mL MCh treatment in control (upper panel) and OVA-induced AR group (lower panel). (**B**) Mucosa index is shown 15 min after PBS or 0.5–2 mg/mL MCh treatment in control (gray bar) and AR (black bar) groups. Data are expressed as means ± SEMs (*n* = 5 for each group). * Significantly different from control mice (*P* < 0.05).

**Figure 4 biomolecules-09-00252-f004:**
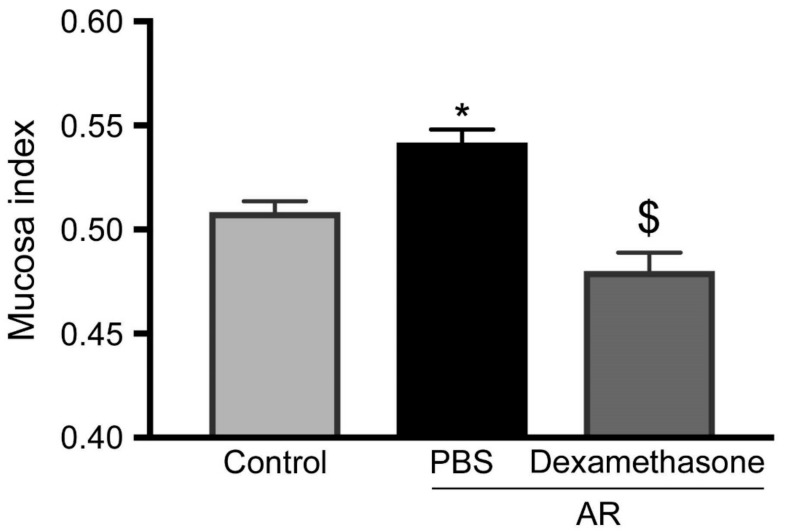
Evaluation of the effect of dexamethasone using micro-CT analysis. Mucosa index in all mice were analyzed 15 min after 1 mg/mL MCh treatment. Light gray, black, and deep gray bars represent healthy controls pretreated with PBS, AR group pretreated with PBS, and AR group pretreated with dexamethasone group, respectively. Data are expressed as means ± SEMs (*n* = 6 for each group). * and $ significantly different from the control and AR groups pretreated with PBS, respectively (*P* < 0.05).

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
