# Peer review of "A Novel Approach for Investigating Upper Airway Hyperresponsiveness Using Micro-CT in Eosinophilic Upper Airway Inflammation such as Allergic Rhinitis Model"

_biomolecules, 2019, doi:10.3390/biom9070252_

Reviewer 1 Report

REVIEWER COMMENT

The authors have submitted a well-conceived manuscript and they mainly focused on an assessment of the upper airway hyperresponsiveness using micro-CT for expanding knowledge in a model of allergic rhinitis. The authors made great efforts in response to their concern using several methods.

In general, the studies are well designed; but it´s essential for readers to describe a scheme protocol more clearly. The data are interesting, appropriately illustrated and accurately referred. Mostly, the experiments are appropriately controlled.

There are a few issues to address and some additional details which would benefit the reader.

Major

1) The mouse model of allergic rhinitis is needed to describe in detail, a scheme protocol is missing therefore is desirable completely to revise this part of method including protocol of sensitization. OVA-induced AR model should be written clear and understandable, it means animals are divided in control group, OVA-group or dexamethasone OVA-group vs. vehicle OVA group (how many animals include each group). Some parts are misleading therefore it is necessary to distinguish between control –naïve animals and control – OVA-animals.

2) To have a complete picture about a model of AR it´s required to describe the complete (behavioural) response of mice on exposure to antigen, which symptoms were observed? What was a variability in response?

3) Reference should be complete in the list of references, i.e. not “Bousquet J et al…” and many others.

4) Because it is an original work, citation of figures is not needed (remove in Fig.1-4 “Bull et al”)

5) The authors advert to Fig. S1-S4 but no such figure occurs in presented work.

Minor

Grammar and typing errors requires a revision on several places.

Line 47: revise prepositions – induced by exposure to inhaled antigens in the upper airway (not from), but in genetically (not genitically)…

Line 76: there is missing is …LAHR is the highest ….

Line 94 – where is Fig. S1A?

Line 99 – where is Fig. S1B?

Line 104, 105, 106 - what was the dose of anesthesia?

Line 107 - …at 0-120 min after intranasal treatment… it requires clarification and describe a direct schedule

Line 143 – correct C on E

Line 168 – maybe “Increasing in”…is better than “Increase in…”

Line 177 – where is Fig.S3?

Line 186 – I suggest more fitting title of subchapter (for example: The effect of steroid treatment on UAHR or Steroids reduced or …)

Line 192 – Fig. 4 there is missing a notion about effect of MCH in the graph

Line 237 – where is Fig. S4?

Line 159 – where is Fig. S2A and S2B?

Line 161 – correct …Analysis of swelling of nasal…, not swollen

Fig. 2 – because pictures in Fig. 2A are quite similar, the authors should point to changes using arrows directly in the picture

Line 167 – Significantly different from PBS-treated…this usage is not usual (example: significantly different between…or significantly different comparing with control group, etc.)

Line 76: there is missing is …LAHR is the highest ….

Author Response

Dear Reviewer1

We found the referees’ comment most helpful and revised above manuscript according comments of reviewer1 (modifications in the text are indicated by red with under line).

Major points

1. The mouse model of allergic rhinitis is needed to describe in detail, a scheme protocol is missing therefore is desirable completely to revise this part of method including protocol of sensitization. OVA-induced AR model should be written clear and understandable, it means animals are divided in control group, OVA-group or dexamethasone OVA-group vs. vehicle OVA group (how many animals include each group). Some parts are misleading therefore it is necessary to distinguish between control –naïve animals and control – OVA-animals.

Answer:  We add a comment in line 100 and modify Fig.S1A.

2) To have a complete picture about a model of AR it´s required to describe the complete (behavioural) response of mice on exposure to antigen, which symptoms were observed? What was a variability in response?

Answer:  Following challenge with specific antigen, OVA-induced AR mice immediately have increasing of sneezing and nose scratching with in 15min, compared with naïve one.    

3) Reference should be complete in the list of references, i.e. not “Bousquet J et al…” and many others.

Answer:  We described references by MDPI format using EndNote. All references would be professionally edited, if the manuscript were to be accepted.

4) Because it is an original work, citation of figures is not needed (remove in Fig.1-4 “Bull et al”)

Answer:  This pre-acceptance format would be also professionally edited, if the manuscript were to be accepted.

5) The authors advert to Fig. S1-S4 but no such figure occurs in presented work.

Answer:  The supplementary material is available for all reviewers. Indeed, another Reviewer mentioned supplementary. Please check it.

In the case of internet trouble, please contact MS Kate Neal who works in the Biomolecules Editorial Office (E-mail: biomolecules@mdpi.com).    

Minor

Line 47: revise prepositions – induced by exposure to inhaled antigens in the upper airway (not from), but in genetically (not genitically)…

Answer:  It is corrected in Line 49.

Line 76: there is missing is …LAHR is the highest ….

Answer:  It is corrected in line 76.

Line 94 – where is Fig. S1A?

Answer:  Please download supplement materials and figures through website of this journal. 

Line 99 – where is Fig. S1B?

Answer:  Please download supplement materials and figures through website of this journal. 

Line 104, 105, 106 - what was the dose of anesthesia?

Answer:  For dose of anesthesia, it is added in line 108-111.    

Line 107 - …at 0-120 min after intranasal treatment… it requires clarification and describe a direct schedule

Answer:  For detail time course, it is added in line 111-114.

Line 143 – correct C on E

Answer:  It is corrected in line 151.

Line 168 – maybe “Increasing in”…is better than “Increase in…”

Answer:  It is corrected in line 176.

Line 177 – where is Fig.S3?

Answer:  Please download supplement materials and figures through website of this journal.

Line 186 – I suggest more fitting title of subchapter (for example: The effect of steroid treatment on UAHR or Steroids reduced or …)

Answer:  We modify it in line 194.

Line 192 – Fig. 4 there is missing a notion about effect of MCH in the graph

Answer:  We already found most effective timing and MCh dose in figure 2 and 3. To investigate the effect of steroid, MCh were administrated in all group under same timing and dose. It was added “in all mice” in line 201-202

Line 237 – where is Fig. S4?

Answer:  Please download supplement materials and figures through website of this journal.

Line 159 – where is Fig. S2A and S2B?

Answer:  Please download supplement materials and figures through website of this journal.

Line 161 – correct …Analysis of swelling of nasal…, not swollen

Answer:  It is corrected in line 169.

Fig. 2 – because pictures in Fig. 2A are quite similar, the authors should point to changes using arrows directly in the picture

Answer:  Arrows are added in Fig.2A.

Line 167 – Significantly different from PBS-treated…this usage is not usual (example: significantly different between…or significantly different comparing with control group, etc.)

Answer:  It is corrected in line 175.

We should like to thank the referee for their helpful comment and trust that the revised manuscript is acceptable for publication in this journal.

Yours sincerely,

Akira KANDA

Reviewer 2 Report

The authors report a  novel approach for investigating upper airway hyperresponsiveness using micro-CT in eosinophilic upper airway inflammation.

In their methodology, I am quite confused with the protocols, the authors may further clarify how they perform the CT scanning especially the time course experiment. For example, were they first anesthesized the mice with ketamine, then killed them with pentobarital and finally to do CT scan in their time course (kinetic) studies. If so, the authors killed different logs of animals at different time point and measure the area of different animals. Since animals within 6-8 weeks may have different head size and I think it is not too scientific to compare the time point difference with different animals. 

So my second question, is it possible to perform the CT scanning using anesthesized animals instead of killed animals? And how long for doing one CT scan? 

The authors present a nice method to quantify the swelling  using the mucosa index, is the software calculating the area highlighted in blue? The authors may mentioned more detailed in the methodology section and figures as well. Also, the index is based on arbitrary values, and it think it may better termed as semi-quantitive.  

Author Response

Dear Reviewer2

We found the referees’ comment most helpful and revised above manuscript according comments of reviewer2 (modifications in the text are indicated by red with under line).

In their methodology, I am quite confused with the protocols, the authors may further clarify how they perform the CT scanning especially the time course experiment. For example, were they first anesthesized the mice with ketamine, then killed them with pentobarital and finally to do CT scan in their time course (kinetic) studies. If so, the authors killed different logs of animals at different time point and measure the area of different animals. Since animals within 6-8 weeks may have different head size and I think it is not too scientific to compare the time point difference with different animals.

Answer:  For detail description about time course, we modify sentence in line 108-114. Regarding of head size, we used mice of weight 20g, selected from 6-8W female (in line 94). Moreover, to correct difference of nasal cavity size, we evaluated swelling of nasal mucus by using M.I. (mucus index) in this study.    

So my second question, is it possible to perform the CT scanning using anesthesized animals instead of killed animals? And how long for doing one CT scan?

Answer:  It is possible in alive mice under anthesis. However, it is quite difficult to take high quality images because of out of focus by slightly moving. For scanning time, it is usually 20 min (it depends on quality of image).

The authors present a nice method to quantify the swelling using the mucosa index, is the software calculating the area highlighted in blue? The authors may mentioned more detailed in the methodology section and figures as well. Also, the index is based on arbitrary values, and it think it may better termed as semi-quantitive. 

Answer:  For this suggestion, we already mentioned it in line 138-141. Briefly, the area colored by blue, indicated in nasal cavity, is automatically captured by OsiriX software following outside drawing of nasal cavity by manual-handling task. For calculation of mucus index (MI), it is calculated as the mean of arbitrary values (slice 1-5.); mucosal area (blue area) / total nasal cavity area (blue and air area).

We should like to thank the referee for their helpful comment and trust that the revised manuscript is acceptable for publication in this journal.

Yours sincerely,

Akira KANDA

Reviewer 3 Report

Allergic rhinitis (AR) is common and there is an urgent need to have better models of disease whereby objective measures of nasal obstruction and nasal hyper-reactivity can be undertaken, for both better understanding of disease mechanisms and to formulate better therapeutic strategies as well as assess therapeutic intervention. This model of a murine model of AR with CT imaging of detailed nasal structures in relation to allergen challenge and provocation with methacholine, showing vascular and inflammation driven mucosal changes is relevant, well presented and interesting.

The introduction is very confusing.

AR is not really called eosinophilic upper airway disease but just allergic rhinitis driven by IgE mediated sensitisation to common aeroallergens with eosinophil recruitment based on the level of disease activation. I would keep the definition of AR and the main mechanism clear. Methacholine is a non-selective muscarinic receptor agonist to stimulate the parasympathetic nervous system. Explain the rationale of using this to provoke AHR.

Nasal hyper-reactivity (sneezing, rhinorrhoea and congestion) is considered the term for upper airway hype-responsiveness and should be clarified as clearly separate to AHR in asthma where airway smooth muscle (ASM)  is present and is the predominant drive for AHR. Vascular oedema, neuronal dysfunction, inflammation are all integrated. Bronchial provocation with methacholine or histamine is the only definite and objective measure of AHR in asthma, and a positive test does not give real insight into the biological pathways that leads to AHR, and currently presumed to be initiated and driven by both inflammation and remodelling. Remodelling in AR is controversial (some authors feel it is not present in a significant manner), my point being AHR mechanisms are complex, and the authors need to present NHR/upper AHR more concisely, lower AHR in asthma and use of methacholine provocation here separately and guide the reader to why this models helps us understand NHR better.

This CT tool is unlikely to be used in the clinical setting as most academic rhinology clinics have access to nasal inspiratory peak flow measurement and acoustic rhinometry to measure nasal airflow resistance objectively, and would not patients to be exposed radiation in this context. The authors need to make the case for why this micro-CT model is helpful i.e. research and drug testing.

Methods

They follow a stand BALB -C mice model of ovalbumin sensitisation and challenge, the time course for testing AHR/NHR when in a chronic inflammatory state is adequate.

As the  Mann-Whitney test requires that the two samples be independent, the methods are appropriate. I am not sure if the small sample sizes are allows sample differences to be detected.

Can the authors comment whether their institutional statistician has advised them on data analysis.

There is no mention of the statistical analysis programme used.

Results

Clearly presented.

Images are well taken.

I am not sure what the immunofluorescence slide in the supplement section adds, and unless a particular cell type correlates with AHR, inflammatory cell data is redundant as the dexamethasone treated group clearly shows that in AR model, inflammation seems to prime the NHR/AHR. 

The duration of NHR/AHR after allergen exposure is currently unknown, and I wonder if the authors have any data on this (in asthma AHR is active even one week after allergen challenge on methacholine testing).

Discussion

Too complex

The points in relation to model relevance and use in the future, insights into NHR mechanism and the key role of inflammation in MCh induced NHS in AR should be the focus. I would not talk about AHR in asthma in details as this is complex, and asthma is not the focus in the paper.  

Author Response

Dear Reviewer3

We found the referees’ comment most helpful and revised above manuscript according comments of reviewer3 (modifications in the text are indicated by red with under line).

Introduction

AR is not really called eosinophilic upper airway disease but just allergic rhinitis driven by IgE mediated sensitisation to common aeroallergens with eosinophil recruitment based on the level of disease activation. I would keep the definition of AR and the main mechanism clear. Methacholine is a non-selective muscarinic receptor agonist to stimulate the parasympathetic nervous system. Explain the rationale of using this to provoke AHR.

The introduction is very confusing.

Nasal hyper-reactivity (sneezing, rhinorrhoea and congestion) is considered the term for upper airway hype-responsiveness and should be clarified as clearly separate to AHR in asthma where airway smooth muscle (ASM) is present and is the predominant drive for AHR. Vascular oedema, neuronal dysfunction, inflammation are all integrated. Bronchial provocation with methacholine or histamine is the only definite and objective measure of AHR in asthma, and a positive test does not give real insight into the biological pathways that leads to AHR, and currently presumed to be initiated and driven by both inflammation and remodelling. Remodelling in AR is controversial (some authors feel it is not present in a significant manner), my point being AHR mechanisms are complex, and the authors need to present NHR/upper AHR more concisely, lower AHR in asthma and use of methacholine provocation here separately and guide the reader to why this models helps us understand NHR better.

Answer:  We agree with comments of reveiwer3 and modify some of introduction.

In addition, AHR reflects epithelial damage with nervous irritability following persistent inflammation and is induced by non-specific factors as well as specific antigen. In other word, AHR does not only reflect specific reaction, and is prominently manifested in the late and silent phage. Although AHR does not give real insight into the biological pathways between upper and lower airway, threshold elevation by MCh model indicates hyperresponsiveness following inflammation including allergic reaction. However, for AHR in the upper airway eosinophilic inflammation mimicking AR, no method has been reported, while evaluation in the early phase by specific antigen has been performed by using frequency of sneezing and scratching.

This CT tool is unlikely to be used in the clinical setting as most academic rhinology clinics have access to nasal inspiratory peak flow measurement and acoustic rhinometry to measure nasal airflow resistance objectively, and would not patients to be exposed radiation in this context. The authors need to make the case for why this micro-CT model is helpful i.e. research and drug testing.

Answer:  We agree with your comments. For now, our novel method is applicable for upper airway inflammation with eosinophilic infiltration model but not human. In the end of this manuscript, we describe an application for effect evaluation of anti-allergic compound in AR by using this system (please see line 256-258).

Methods

They follow a stand BALB -C mice model of ovalbumin sensitisation and challenge, the time course for testing AHR/NHR when in a chronic inflammatory state is adequate.

As the Mann-Whitney test requires that the two samples be independent, the methods are appropriate. I am not sure if the small sample sizes are allows sample differences to be detected.

Can the authors comment whether their institutional statistician has advised them on data analysis.

There is no mention of the statistical analysis programme used.

Answer:  Because samples were 5-6, we used Mann–Whitney U test. For software of statistical analysis, we describe it in line 127  

Results

I am not sure what the immunofluorescence slide in the supplement section adds, and unless a particular cell type correlates with AHR, inflammatory cell data is redundant as the dexamethasone treated group clearly shows that in AR model, inflammation seems to prime the NHR/AHR.

The duration of NHR/AHR after allergen exposure is currently unknown, and I wonder if the authors have any data on this (in asthma AHR is active even one week after allergen challenge on methacholine testing).

Answer:  In the Fig.S1, we showed that SiglecF positive cells, indicated eosinophils, were infiltrated into nasal mucosa in our experimental model mimic allergic rhinitis, because eosinophil-derived granules such as cationic proteins contribute to development of AHR (Pulm Pharmacol Ther. 2019 May 13:101804. doi: 10.1016).  

For duration, yes, it is very interesting point. Unfortunately, we did not investigate it and were supposed to do that. 

Discussion

Too complex

The points in relation to model relevance and use in the future, insights into NHR mechanism and the key role of inflammation in MCh induced NHS in AR should be the focus. I would not talk about AHR in asthma in details as this is complex, and asthma is not the focus in the paper. 

Answer:  We also agree with this reviewer’s comment. In the measurement of LAHR, it has frequently been evaluated by airway resistance, consisted with smooth muscle constriction. By contrast, studies about UAHR evaluated by nasal resistance has hardly reported, because nasal resistance consists with nasal discharge and swelling of nasal mucosa. For this reason, in this study, we would like to discuss about difference between lower and upper airway AHR but not detail mechanisms of UAHR, and compare the advantage to disadvantage of micro-CT with another approach such as whole-body plethysmography. Please see Fig.S4.

Notably, for nasal resistance, we already developed novel measurement system for upper airway resistance and then found neural pathway in the upper airway (unpublished data). Moreover, we already show that anticholinergic compound prevents swelling of nasal mucosa in AR model (unpublished data). These data will be reported near future. 

Round  2

Reviewer 1 Report

The quality of revised manuscript has been improved with regard to reviewer’s comments and suggestions. This revised manuscript still requires minor correction of some teeny, inconsiderable stylistic and grammar inaccuracy.

Reviewer 2 Report

No further comments

Reviewer 3 Report

Please correct some grammar and spelling errors.